# Electron cryo-tomography structure of axonemal doublet microtubule from *Tetrahymena thermophila*

Sam Li[1] , Jose-Jesus Fernandez[2] , Amy S Fabritius[3], David A Agard[1] , Mark Winey[3]

**Doublet microtubules (DMTs) provide a scaffold for axoneme assembly in motile cilia. Aside from $\alpha/\beta$ tubulins, the DMT comprises a large number of non-tubulin proteins in the luminal wall of DMTs, collectively named the microtubule inner proteins (MIPs). We used cryoET to study axoneme DMT isolated from *Tetrahymena*. We present the structures of DMT at nanometer and sub-nanometer resolution. The structures confirm that MIP RIB72A/B binds to the luminal wall of DMT by multiple DM10 domains. We found FAP115, an MIP-containing multiple EF-hand domains, located at the interface of four-tubulin dimers in the lumen of A-tubule. It contacts both lateral and longitudinal tubulin interfaces and playing a critical role in DMT stability. We observed substantial structure heterogeneity in DMT in an *FAP115* knockout strain, showing extensive structural defects beyond the FAP115-binding site. The defects propagate along the axoneme. Finally, by comparing DMT structures from *Tetrahymena* and *Chlamydomonas*, we have identified a number of conserved MIPs as well as MIPs that are unique to each organism. This conservation and diversity of the DMT structures might be linked to their specific functions. Our work provides structural insights essential for understanding the roles of MIPs during motile cilium assembly and function, as well as their relationships to human ciliopathies.**

## Introduction

The motile cilium is an evolutionarily conserved cellular organelle that has a wide range of biological functions. In protozoa, such as ciliates and green algae, motile cilia, also known as flagella, are essential for the cell's locomotion and feeding (Bornens, 2018). In metazoa, motile cilia have multiple functions. For example, in mammals, they are concentrated in the airway for mucus clearance, and in the ventricular tube of the brain for cerebrospinal fluid flow. In addition, motile cilia are found in the oviduct of

female reproduction organs, in the male sperm and at the node of the gastrulation site during embryo development. Defective motile cilia cause a spectrum of human diseases, such as primary cilia dyskinesia, situs inversus, and infertility (Reiter & Leroux, 2017).

In the core of a motile cilium, two singlet microtubules, known as the central pair, together with nine doublet microtubules (DMTs) that are radially surrounding the central pair, establish a 9+2 configuration, a hallmark of the motile cilium. These microtubule-based structures provide a scaffold for assembly of hundreds of other ciliary components, such as the outer/inner dynein arms, the dynein regulatory complex, and the radial spokes, all of which are essential for driving and regulating cilia beating. Fueled by ATP, the DMT-associated dynein generates power strokes that effectively slide one DMT relative to its adjacent DMTs. The continuous and asynchronous alternation of power stroke and sliding of DMTs on the opposite sides of the axoneme translates into a local bending of the cilium. This bending propagates along the cilium length manifested as various waveforms during ciliary beating (Mitchison & Mitchison, 2010; Lin & Nicastro, 2018).

In addition to having an essential scaffold role during cilia beating, the DMT also serves as a bi-directional track for the intraflagellar transport important for cilia assembly and maintenance (Rosenbaum & Witman, 2002). Kinesin motors drive anterograde transport moving on the B-tubule towards the plus end, whereas the cytosolic dyneins move in the opposite direction on the A-tubule for retrograde transport (Stepanek & Pigino, 2016).

The DMT comprises a full 13-protofilament (pf) A-tubule and an incomplete 10-pf B-tubule that shares a wall, known as the ribbon region, with the A-tubule. Besides $\alpha/\beta$ tubulins, it has a large number of proteins decorating the luminal wall of the A- and B-tubule, known as the microtubule inner proteins (MIPs) (Sui & Downing, 2006; Nicastro et al, 2006, 2011). Recent structure studies, in particular by cryoEM single particle analysis and tomography (cryoET), have revealed the molecular architecture of the DMT in great details (Ichikawa et al, 2017, 2019; Ma et al, 2019; Khalifa et al, 2020; Song et al, 2020). These results showed that the MIPs non-uniformly decorate the luminal wall of the DMT. They form a tightly knitted, highly intricate meshwork bolstering the MT lattice. The structures highlighted the complexity of motile cilia as functional

[1]Department of Biochemistry and Biophysics, University of California, San Francisco, CA, USA   [2]Nanomaterials and Nanotechnology Research Center (CINN-CSIC), Health Research Institute of Asturias (ISPA), Oviedo, Spain   [3]Department of Molecular and Cellular Biology, University of California Davis, Davis, CA, USA

Correspondence: sam.li@ucsf.edu; mwiney@ucdavis.edu

machinery and have substantially advanced our understanding of the composition of the axoneme in the context of its 3D structure. Meanwhile, these findings have raised many questions. For example, it is unclear precisely how the MIPs contribute to the function of motile cilia. Also, mechanistic detail on how these large numbers of MIPs are coordinated and their inter-dependence during the DMT assembly process is largely unknown. In addition, the structural conservation and diversity of the DMT in other organisms is unclear. Here, we address many of these questions by reporting the cryoET results on the axonemal DMT isolated from *Tetrahymena thermophila*. In particular, we focused our analysis on the MIPs in the luminal wall of A- and B-tubules.

## Results

### Overall structure of the DMT from *T. thermophila*

Previous studies show the MIPs decorate the *Tetrahymena* DMT wall asymmetrically, manifesting a range of longitudinal periodicities, from 8 nm up to 48 nm (Ichikawa et al, 2019; Song et al, 2020). To survey the entire structure at the highest possible resolution, we initially focused on the MIPs with 16 nm periodicity. By aligning the A- and B-tubule separately, we obtained averaged maps of the DMT in an overall resolution of 10.2 Å (Figs 1A and S1 and Table 1). In the average, in particular the MT backbones, the secondary structural elements can be resolved (Fig 1B). Meanwhile, many non-tubulin proteins on the luminal wall of DMT can be discerned based on their overall shapes. These include RIB72 and FAP115 in the lumen of A-tubule, FAP20, and PACRG forming a filament at the inner

junction, and FAP52, whose two WD40 domains each with characteristic seven β-propeller repeats can be resolved (Fig 1C). Meanwhile, a number of filamentous MIPs (fMIPs) associated at either inside or outside cleft between MT protofilaments can be resolved. Overall, the improved resolution facilitates identification of the MIPs in the DMT.

### RIB72A/B on the luminal wall of A-tubule

The conserved RIB72 protein is an elongated multidomain MIP essential for flagella function and for recruiting other MIPs to the organelle (Stoddard et al, 2018; Ma et al, 2019; Fabritius et al, 2021). A human homolog of RIB72, EFHC1, is linked to juvenile myoclonic epilepsy (Suzuki et al, 2004, 2020; Gonsales et al, 2020). Based on the atomic model of RIB72 from *Chlamydomonas*, its N terminus starts at pf A12 in the ribbon region. The molecule spans seven pfs in the luminal wall of the A-tubule. Its C-terminal domain (CTD) anchors at pf A05. Whereas *Chlamydomonas* has a single copy of the *RIB72* gene, *Tetrahymena*, like many other ciliates, possesses two paralog genes, *RIB72A* and *RIB72B*, whose corresponding proteins share 41% identity. Both proteins have three tandem DM10 domains connected by flexible linkers. However, RIB72B lacks a C-terminal EF-hand motif. Previous studies showed that single or double knockouts of *RIB72A* and *RIB72B* in *Tetrahymena* caused extensive defects in the DMT and reduced cell motility (Stoddard et al, 2018). Here, we further investigated the binding of RIB72A/B to the MT wall and to other MIPs. By focusing subtomogram alignment on the pf A01~A06 region, the structure was improved slightly from 10.2 to 9.9 Å (Figs 2A and S2 and Table 1). In the average, RIB72A and B are distinguishable, alternating longitudinally every 8 nm. In each

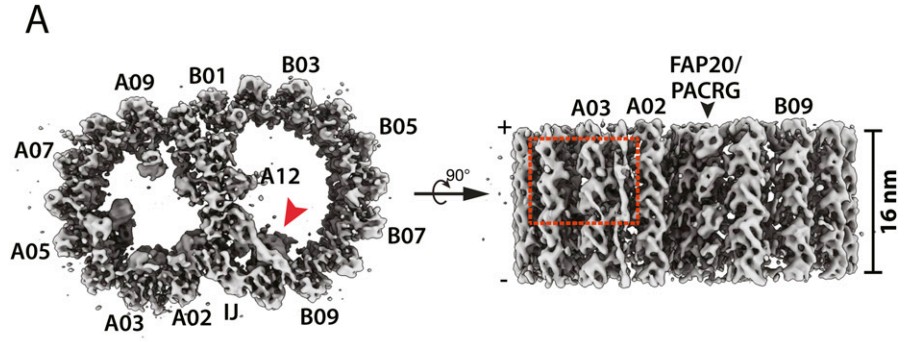

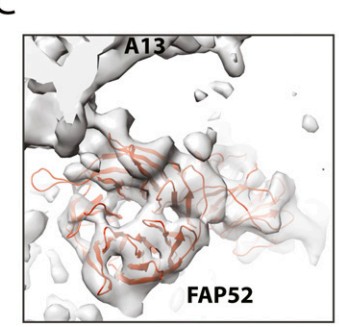

**Figure 1. Structure of the 16-nm repeat of the doublet microtubule from the wild type.**
**(A)** The DMT structure is displayed as iso-surface representation in two orthogonal views. Left: the DMT is viewed from the plus end. IJ: inner junction. Right: the DMT is viewed from the lumen of axoneme. **(B)** A magnified view of the boxed region in (A) shows that the secondary structure elements can be resolved in many parts of the map. Atomic models of α (green) and β (light blue) tubulin are fit into the protofilament A03 in the map. **(C)** An atomic model of FAP52, one of the microtubule inner proteins at the inner junction whose location in the DMT is indicated by a red arrowhead in (A), is fit into the average map. The blades of the β-propeller fold from the FAP52 WD40 domain are resolved in the map.

**Table 1. Summary of Structure from wild type and *FAP115KO* mutant.**

| Structure | Number of subtomogram | Resolution (Å) | Description | EMDB ID # |
|---|---|---|---|---|
| 1 | 16,125 | 10.2 | A-tubule, 16 nm, wild type | EMD-24364 |
| 2 | 15,033 | 10.2 | B-tubule, 16 nm, wild type | EMD-24366 |
| 3 | 13,034 | 9.9 | Partial A-tubule, 16 nm, focusing on RIB72A/B, wild type | EMD-24367 |
| 4 | 5,083 | 12.0 | A-tubule, 48 nm, wild type | EMD-24368 |
| 5 | 5,855 | 12.0 | B-tubule, 48 nm, wild-type | EMD-24370 |
| 6 | 16,967 | 9.6 | A-tubule, 16 nm, *FAP115KO* | EMD-24371 |
| 7 | 17,133 | 9.6 | B-tubule, 16 nm, *FAP115KO* | EMD-24372 |
| 8 | 12,697 | 9.6 | Partial A-tubule, 16 nm, focusing on RIB72A/B, *FAP115KO* | EMD-24373 |
| 9 | 9,502 | 11.8 | A-tubule, 48 nm, *FAP115KO* | EMD-24374 |
| 10 | 9,502 | 12.0 | B-tubule, 48 nm, *FAP115KO* | EMD-24375 |
| 11 | 4,960 | 12.0 | An unknown MIP in inner junction region, 48 nm | EMD-24376 |
| 12 | 4,960 | 10.9 | Focusing on an unknown MIP in inner junction region, 24 nm | EMD-24377 |
| 13 | — | — | Composite map combining #1 #2, 16 nm, wild type | EMD-24379 |
| 14 | — | — | Composite map combining #4 #5, 48 nm, wild-type | EMD-24380 |
| 15 | — | — | Composite map combining #6 #7, 16 nm, *FAP115KO* | EMD-24381 |
| 16 | — | — | Composite map combining #9 #10, 48 nm, *FAP115KO* | EMD-24382 |

RIB72A or B, three DM10 domains can be readily recognized binding to pfs A01, A03, and A05 respectively. We fit an atomic model of RIB72 from *Chlamydomonas* into the average map (Fig 2A). For RIB72B, the last 84 residues were removed, corresponding to the EF-hand motif. The models fit into the average map without any local modification, indicating that the topology of RIB72, including the DM10 domains and the length and curvature of corresponding linkers, are conserved between these two organisms (Fig 2A). This is also consistent with the previous observation that the sequences of each DM10 domain are more highly conserved between proteins from different species than they are within a particular molecule (King, 2006), suggesting each domain has unique and conserved functions.

Like *Chlamydomonas*, each RIB72A and B uses three DM10 domains to anchor to the DMT wall. We built pseudo-atomic models for the corresponding pfs of the DMT. In the model, each DM10 domain binds at the longitudinal α/β tubulin interdimer interface. In the average, five out of six DM10 domains from RIB72A/B showed connecting density between the DM10 domain and the α-tubulin (Fig 2B). Based on the models, these connections are between the last α-helix in the DM10 domains and the H1-S2 loop in α-tubulin, also known as the αK40 loop. The loop is disordered in the undecorated MT (Zhang et al, 2015; Howes et al, 2017; Eshun-Wilson et al, 2019). Interestingly, the αK40 residue is a site for post-translation modification by acetylation (Akella et al, 2010). Although at current resolution we could not resolve the lysine residue or the acetyl moiety in the structure, our observations corroborate the recent observations of the RIB72 in *Chlamydomonas* (Ma et al, 2019). The observed connecting densities between the DM10 domains and the αK40-loops suggest that the lysine (K40) residues in these α tubulins are acetylated. The RIB72A/B DM10 domains recognize and bind at the longitudinal interface of α/β tubulin at pfs A01, A03, and A05, and it is likely that one of the recognition

features is the acetylated αK40. Intriguingly, although the K40 acetylation is not essential for survival of *Tetrahymena*, depletion of K40 acetylation resulted in elevated MT dynamics (Gaertig et al, 1995; Akella et al, 2010). In addition to a direct mechanical role that the acetylated α-tubulin might play by stabilizing the MT lattice during its bending (Xu et al, 2017; Eshun-Wilson et al, 2019), it is likely that the acetyl-K40 could also provide a binding site for recruiting MIPs, which will in turn further stabilize the DMT.

In *Chlamydomonas*, the two longitudinally neighboring RIB72s are crosslinked at RIB72's N- and C terminus by the MIPs FAP222 and FAP252, respectively (Ma et al, 2019). However, we could not find corresponding densities in the *Tetrahymena* structure. Instead, we identified a previously unreported MIP at pf A04~A05 that is unique to *Tetrahymena* (Fig 2C and Videos 1 and 2). The MIP has an elongated bi-lobe shape, repeating every 16 nm in the longitudinal direction. It crosslinks the neighboring RIB72A and B in the luminal side of pf A05. At one end, a large globular domain contacts the EF-hand motif of RIB72A. A small domain directly binds to the DM10-3 domain of RIB72B. The two globular domains are connected by a funnel-shaped linker. Meanwhile, the MIP also makes multiple interactions with α/β tubulin dimer at pf A04 and with the linker region of RIB72A (Fig 2C). The estimated molecular weight of this MIP is ~27 kD.

### Binding of FAP115 on the luminal wall of A-tubule

Recent studies showed that FAP115, a protein with multiple EF-hand motifs, was one of the MIPs missing in the *rib72* null mutations, suggesting the recruitment of FAP115 to the axoneme depends on RIB72 (Stoddard et al, 2018; Ma et al, 2019). Conversely, a *Tetrahymena FAP115* knockout mutant retained RIB72A/B, suggesting RIB72's binding to the axoneme is independent of FAP115 (Fabritius et al, 2021). To further understand the molecular mechanism of

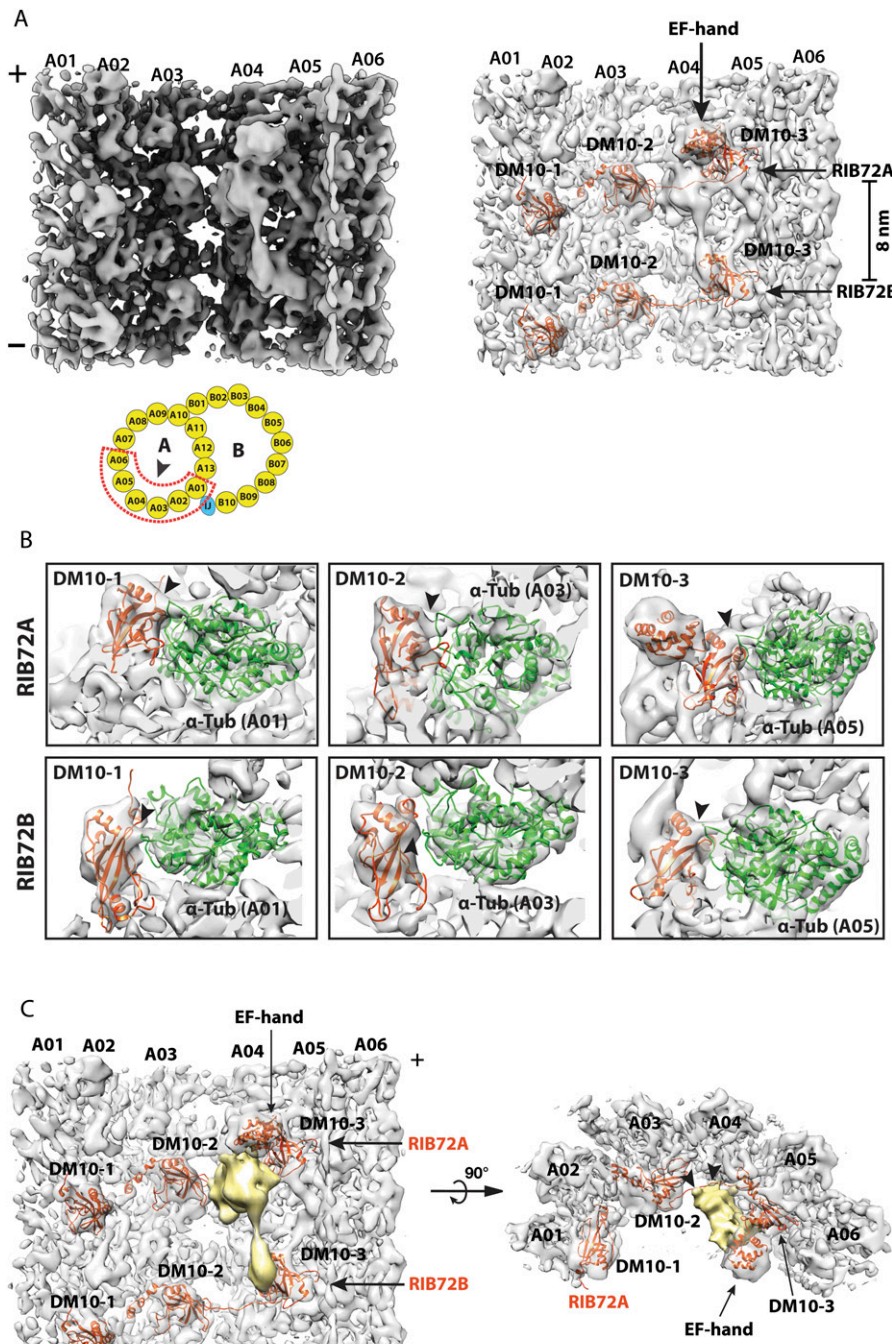

**Figure 2. Structure of the 16-nm repeat of the doublet microtubule focusing on the RIB72A and RIB72B binding region.**
**(A)** A side view of the density map focusing on the RIB72A/B binding sites. Left: the map is viewed from the luminal side of DMT as indicated by an arrowhead in the cartoon. Right: pseudo-atomic models of RIB72A and RIB72B, based on the structure from *Chlamydomonas*, are fit into the density map. The six DM10 domains and an EF-hand motif in the RIB72A C terminus are indicated. In the cartoon, the red dashed lines outline the structure feature used for focused local refinement. This includes the protofilaments A01~A06 and their associated microtubule inner proteins (MIPs). **(B)** The DM10 domains bind to the K40 loops at the luminal side of α-tubulin. The models for the DM10 domains (in red) are fit into the density map. The α-tubulins are in green. The black arrowheads indicate the connecting densities and the potential interactions between the DM10 domain and the K40 loop from α-tubulin. The top row shows three DM10 domains from RIB72A. The bottom row shows three DM10 domains from RIB72B. **(C)** An unidentified MIP (in gold) crosslinks the C terminus of RIB72A and RIB72B. The averaged density map is shown in two orthogonal views. The black arrowheads indicate the binding sites of the unidentified MIP to the linker between DM10-2 and DM10-3 from RIB72A and to protofilament A04.

FAP115 function, we used cryoET to study FAP115 in both wild-type and a *FAP115* knockout mutant. In the wild-type, FAP115 binds to pf A02~A03, repeating every 8 nm in the longitudinal direction. Based on the models from *Chlamydomonas*, we built pseudo-atomic models of FAP115 and its neighboring RIB72A/B (Fig 3A). The model of FAP115 fits nicely into the map as shown by its secondary structure elements that could be clearly resolved. In our structure, FAP115 is shown as a bilobed structure with two domains, the N-terminal domain (NTD) and the C-terminal domain (CTD). They were connected by a flexible linker. Each domain has two EF-hands. The linker has an extended loop reaching out in the DMT plus-end direction making potential contact with the DM10-1 domain of RIB72 (Fig 3A). The NTD of FAP115 was anchored at the longitudinal interface of α/β tubulin heterodimer on pf A02 (the N-site, the non-exchangeable GTP site), potentially contacting both tubulins. The FAP115 CTD is particularly interesting. It is located at the junction of four-tubulin dimers interface, potentially interacting with all four tubulins (Fig 3B). It fits into an ideal space sandwiched between two

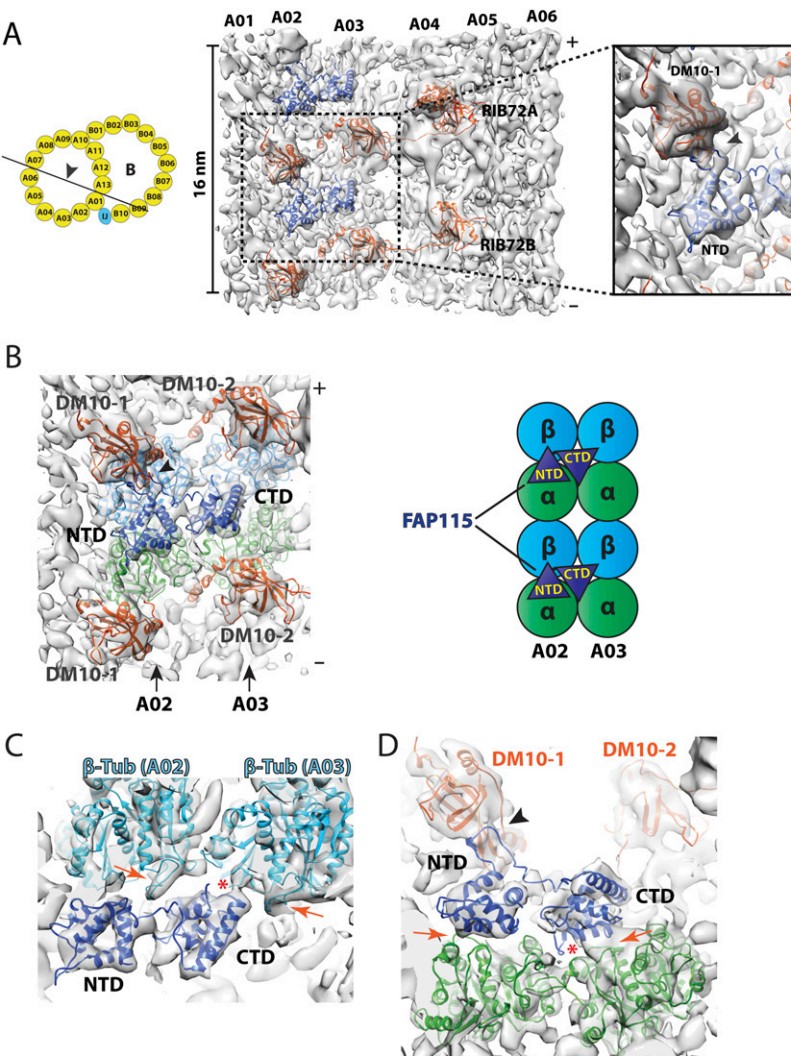

**Figure 3. Binding of FAP115 to the luminal wall of doublet microtubule.**
**(A)** FAP115 binds to protofilament A02 and A03 in the lumen of A-tubule. The atomic models for FAP115 (dark blue) and RIB72A/B (red) are fit into the density map. The viewing direction of the map is depicted in a cartoon on the left. The inset on the right shows a magnified view of FAP115. An arrowhead in the inset indicates the potential interaction between an extended loop from FAP115 and the first DM10 domain (DM10-1) from RIB72A that is anchored at protofilament A01. **(B)** Binding of FAP115 to protofilament A02 and A03. The density map is shown in the same viewing direction as in (A). Two pairs of α/β tubulin dimers are fit into the density map, α-tubulins are in green and β-tubulins are in light blue. The two EF-hand domains from FAP115, the N-terminal domain (NTD) and the C-terminal domain (CTD), are in dark blue. The cartoon illustrates the binding of FAP115 to two pairs of α/β tubulin dimers from pf A02/A03. **(C)** Potential interactions between FAP115 and two β-tubulins from pf A02/A03. The H1S2 loops from two β-tubulins are indicated by the red arrows. These two loops resemble two jaws of a vernier caliper where FAP115 CTD fits in and makes contacts. A red asterisk indicates potential interaction between FAP115 CTD and the S9-S10 loop from A03 β-tubulin, Cα-Cα distance < 7 Å. **(D)** Potential interactions between FAP115 and two α-tubulins from pf A02/A03. The two red arrows indicate potential interaction interface between FAP115 NTD, CTD and two α-tubulins, Cα–Cα distance < 7 Å. A red asterisk indicates an EF-hand loop from FAP115 that extends towards the lateral interface between two α-tubulins, making potential interactions.

β-tubulins from pf A02 and A03 (Fig 3C and Video 3). Specifically, the two H1-S2 loops from these β-tubulins resemble two jaws of a vernier caliper that contact the FAP115 CTD. Meanwhile, the S9-S10 loop from the A03 β-tubulin also interacts with the CTD. In addition to the two β-tubulins, the FAP115 CTD also makes several interactions with two α-tubulins from pf A02 and A03. Notably, a loop of the EF-hand (EF-loop) extends towards the tubulin lateral interface, making potential contacts with the H1-S2/H2-S3 loops from A02 α-tubulin and the M-loop from A03 α-tubulin (Fig 3D and Video 4). Because of the limited resolution in our structure, we could not directly observe side chain interactions; therefore, caution has to be taken when interpreting the model. Nevertheless, the observed binding of FAP115 CTD at the interface of four tubulin dimers is reminiscent of the binding of the calponin-homology (CH) domain of the MT plus-end tracking EB proteins or the DC domain of doublecortin, an MT-stabilizing protein expressed in developing neurons. All these MAP domains preferentially bind to 13-pf MT and make contact to four neighboring tubulins on the outer surface of MT (Fourniol et al, 2010; Maurer et al, 2012; Zhang et al, 2015).

Although the details of the interaction are completely different, our observation indicated that FAP115 used a similar strategy as EBs or doublecortin for binding and stabilizing both longitudinal and lateral interfaces of MT lattice, specifically at the pf A02 and A03 interfaces, to help strengthen the DMT.

### FAP115 knockout results in axoneme defects beyond its binding site

To further understand the function of FAP115 in motile cilia beating, we studied axonemes isolated from a *FAP115* knockout strain (*FAP115KO*). Because FAP115 exhibits 8-nm longitudinal periodicity in the wild type, to improve the resolution in the average, we limited the mutant axoneme length to 18 nm containing two repeats. We further focused the alignment on the pf A01~A06 region where the RIB72A/B and the FAP115 bind in the wild-type. The average of the *FAP115KO* mutant reached 9.6 Å in resolution (Figs 4A and S3A and Table 1). Compared with the wild type, the bi-lobed structure of FAP115 is absent in the mutant (Figs 4B and S3B), confirming the

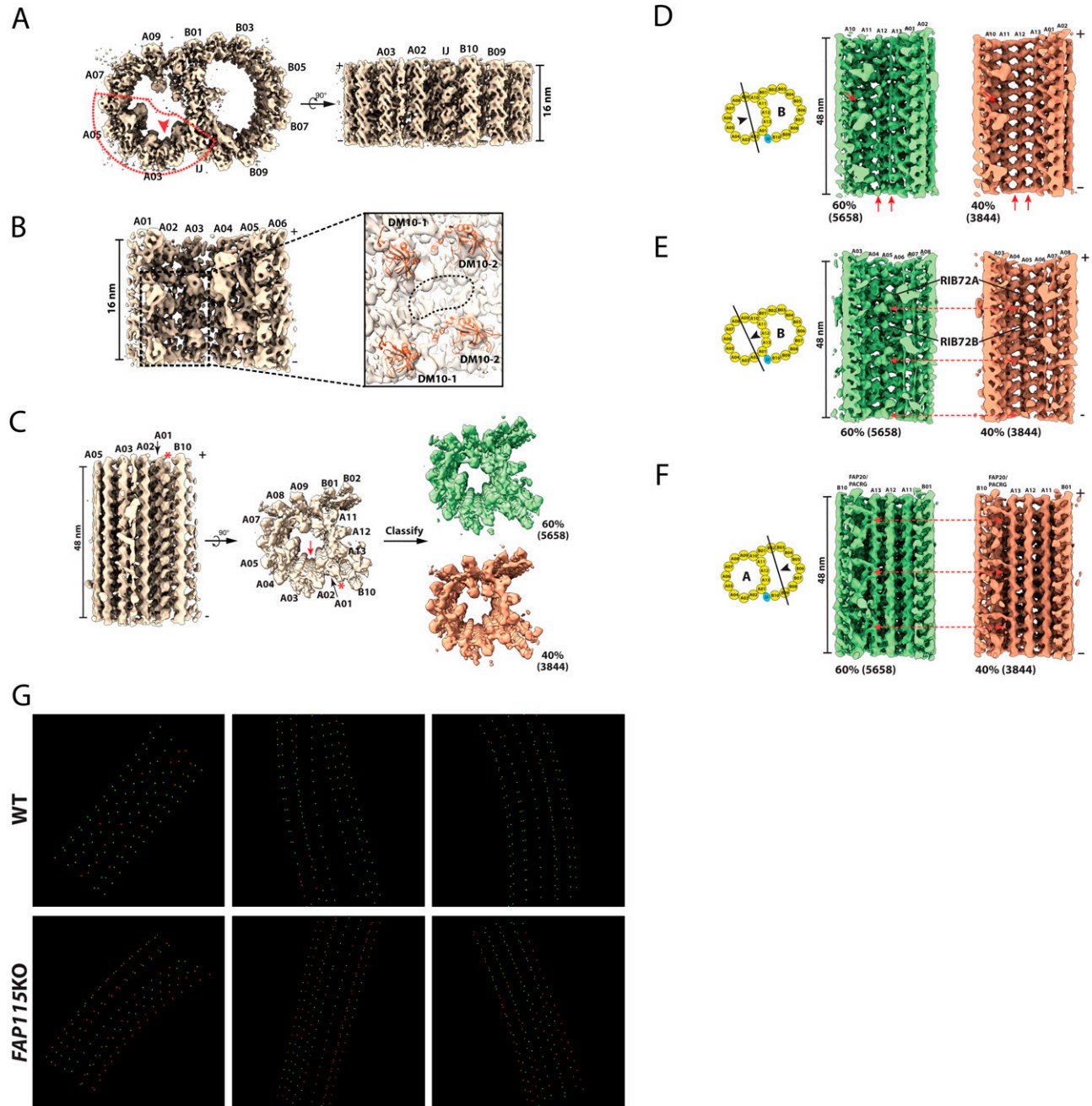

**Figure 4. Structures of the doublet microtubule from *FAP115* knockout mutant.**
**(A)** Structure of the 16-nm repeat of the doublet microtubule from the *FAP115KO* mutant. Two orthogonal views of the structure are shown. A red arrowhead indicates the location of FAP115 in the wild type and the viewing direction in (B). The red dashed lines outline the region of A-tubule centered on FAP115 that was used in the focused refinement. The refinement result is shown in (B). **(B)** Structure of part of A-tubule from *FAP115KO* mutant viewed from the lumen. The inset shows a magnified view of the boxed area. The models for RIB72A/B (red) are fit into the density map. The DM10 domains are indicated. The dashed oval circle highlights the location of FAP115 in the wild type, but it is empty in the *FAP115KO* mutant structure. **(C)** Classification of the 48-nm Repeat of the A-tubule from the *FAP115KO* mutant. The structure is shown in two orthogonal views. A red arrow indicates the expected location of FAP115 in the wild type. A red asterisk indicates the inner junction protofilament composed of PACRG and FAP20. The classification focusing on the ribbon region pf A11~A13 resulted in two classes as shown on the right – the 60% intact class in green and the 40% defective class in orange. The number of subtomograms in each class is indicated in parenthesis. **(D, E, F)** Comparison of the two class-averages reveals major structure differences in three regions of the DMT, the ribbon region, the A04A05 region and the inner junction region. The viewing directions are depicted by the black arrowheads in the cartoons. The two class-averages are displayed side-by-side. The red arrows highlight the structure differences in each region. **(G)** Mapping the intact and the defective DMT structures in the axoneme. Three representative tomograms from the wild-type and the *FAP115KO* mutant are shown. Each green or red point represents a 48-nm segment of the DMT, green: intact structure. red: defective structure. The continuous dotted line represents the DMT. Each dataset has nine dotted lines representing nine DMTs in the axoneme.

depletion of FAP115 in the mutant (Fabritius et al, 2021). However, RIB72A/B remains in the *FAP115KO* mutant structure (Fig 4B). This is in contrast to the *rib72* null mutants where the FAP115 was missing (Stoddard et al, 2018; Ma et al, 2019), suggesting the recruitment of RIB72A/B to the axoneme is independent of FAP115. Besides the absence of FAP115, we did not observe other defects in the mutant structure. For example, despite the FAP115 crosslinking of pf A02~A03 in the wild-type, we did not detect a change of curvature between pf A02 and A03 in the mutant. It is possible that the curvature or other structural change is subtle in the *FAP115KO* mutant and our current resolution is not high enough to detect the change. Alternatively, the DMT lattice is a robust assembly, and other MIPs might have overlapped function with the FAP115 in maintaining the DMT structural integrity.

We speculated that limiting our DMT segment length to 18 nm may have resulted in MIPs with longer periodicity, such as 48 nm, being averaged out. To overcome this, we extended the segment length to 53 nm in subtomogram averaging, for both the wild-type and the *FAP115KO* (Fig S3G and H and Table 1). We further classified the averages, focusing on the ribbon region spanning pf A11~A13. For the *FAP115KO* mutant, we obtained two distinct classes from 9,502 subtomograms, representing 60% (5,658 subtomograms) and 40% (3,844 subtomograms) of the population, respectively (Fig 4C). Surprisingly, although the classification was restrained to the ribbon region, careful examination of these two classes revealed substantial structural differences extending beyond the ribbon region. First, there is a substantial loss of MIPs in the ribbon region in the 40% class (Fig 4D). Most notable is the loss of RIB43a-S/L, two fMIPs running longitudinally along the inner clefts between pf A11~A12 and A12~A13, respectively in the wild-type (Ichikawa et al, 2019). In addition, a number of MIPs in the ribbon region, presumably associated with RIB43a-S/L, were also missing in this 40% class. Another structural difference is in the pf A04~A05 region. The 40% class has substantial reduction of the aforementioned MIP density that longitudinally crosslinks RIB72A/B in the wild-type (Figs 4E and 2C). In contrast, the RIB72A/B remained present in both classes. Finally, a "tether density" at the inner junction is missing in the 40% class (Fig 4F). To rule out the possibility that these observed structural differences were caused by any artifact introduced during axoneme isolation, we examined the wild-type structure. The classification identified 772 (13%) from 5,855 subtomograms in the wild type having defects in the ribbon region (Fig S3C and D). However, a close inspection on this minor class showed that other regions in DMT, including the pf A04~A05 and the inner junction region remain intact (Fig S4E and F). This is in contrast to the observation in the 40% class from the *FAP115KO* mutant where structural defects were identified in multiple locations. Taken together, this analysis demonstrated that the structural defects observed in the 40% of *FAP115KO* mutant DMT were the consequence of the depletion of FAP115. It revealed that defects in the DMT due to the loss of FAP115 extended beyond its binding site at pf A02~A03.

The RIB43a-S/L are fMIPs in the ribbon region connected repeatedly in a head-to-tail fashion, forming two continuous filaments threading longitudinally throughout the axoneme. It is likely they provide a scaffold for other MIPs, binding and strengthening the local MT lattice. Because one of the major defects observed in

the *FAP115KO* was the loss of RIB43a-S/L and other associated MIPs in the 40% fraction of DMT, we asked if the defects would have any global impact on the axoneme structure. To address this question, we mapped the defective DMT identified by classification to the axoneme tomograms. The mapping of the two classes shows the defects were clustered in the mutant axoneme. Many formed long continuous stretches of defective DMT (Fig 4G). Given that many MIPs, in particular, the fMIPs, such as RIB43a-S/L, form interwoven meshwork in the lumen of DMT, this is likely the result of a "domino effect" where a local structural defect affects its neighbor by propagating the abnormality along the DMT. In contrast to the mutant, the defective DMT detected in wild type were dispersed and restricted to local sites in the axoneme as shown in the tomograms (Fig 4G). In summary, we revealed that depletion of FAP115 caused substantial defective DMT in the axoneme. The defect was not limited to the FAP115-binding site, instead, it spread to several regions in DMT. Furthermore, the defects were propagated along the DMT in axonemes of the *FAP115KO* mutant.

### Structural comparison of MIPs in the DMT between *Tetrahymena* and *Chlamydomonas*

The motile cilium is an evolutionarily conserved organelle (Satir et al, 2008; Mitchell, 2016). However, cilia from different organisms or species exhibit unique characteristics, ranging from differences in length, waveform, beating frequency and amplitude, and in response to various external stimuli. These differences are likely underlain by their unique molecular composition and structure (Satir et al, 2008). Previous studies have revealed marked differences in the cilia structure among different model organisms (Carvalho-Santos et al, 2011; Nicastro et al, 2011; Pigino et al, 2012; Lin et al, 2014; Khalifa et al, 2020). However, many comparisons were descriptive and were limited by the resolution of the structures. Recently, the structure of DMT in *Chlamydomonas* has been solved at near atomic resolution (Ma et al, 2019). In the light of these revelations, we sought to compare our DMT structure from *Tetrahymena* to the structure in *Chlamydomonas*, focusing on the MIPs. The comparison was not meant to be exhaustive, but by taking a heuristic approach we intend to learn the structure conservation and diversity of the DMT between these two model organisms that will infer their functions.

Because the binding of MIPs to the DMT is not uniform, besides the aforementioned structural difference in the pf A01~A06 region, we chose to compare three additional regions clustered with MIPs. For each region, we took a set of MIP models from *Chlamydomonas*, used their relative distance and orientation as constraints and fit the models into our 48 nm DMT structure from the wild-type.

First, we compared the A-tubule seam region spanning pf A06~A10. This region is populated by MIPs with 48-nm periodicity. We found two fMIPs in our map, corresponding to FAP53 and FAP127 from *Chlamydomonas* (Fig 5A). The models fit into our map very well, including the local kinks and bulges of the filaments. In particular, both FAP53 and FAP127 turned 90 degrees at the expected positions, transitioning from longitudinal to transverse direction across the seam towards the outer junction, indicating their high degree of structural conservation. In addition, we also found two copies of FAP67, a nucleoside diphosphate kinase.

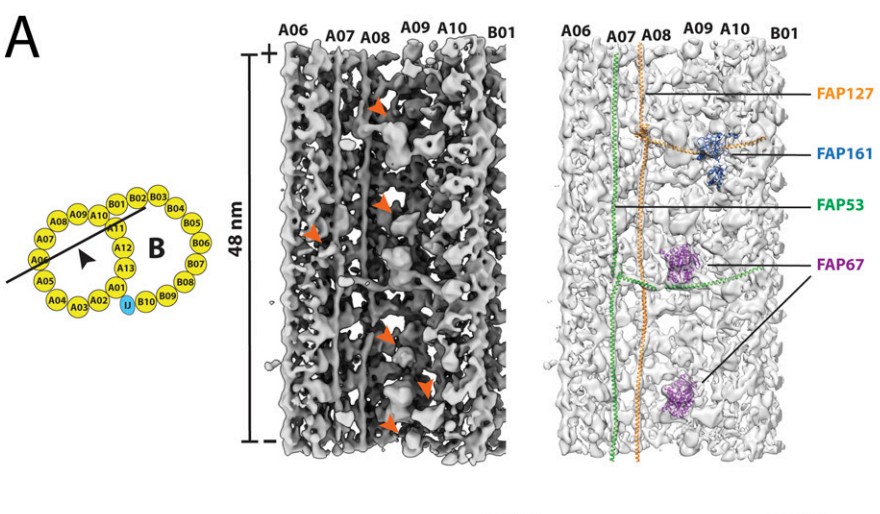

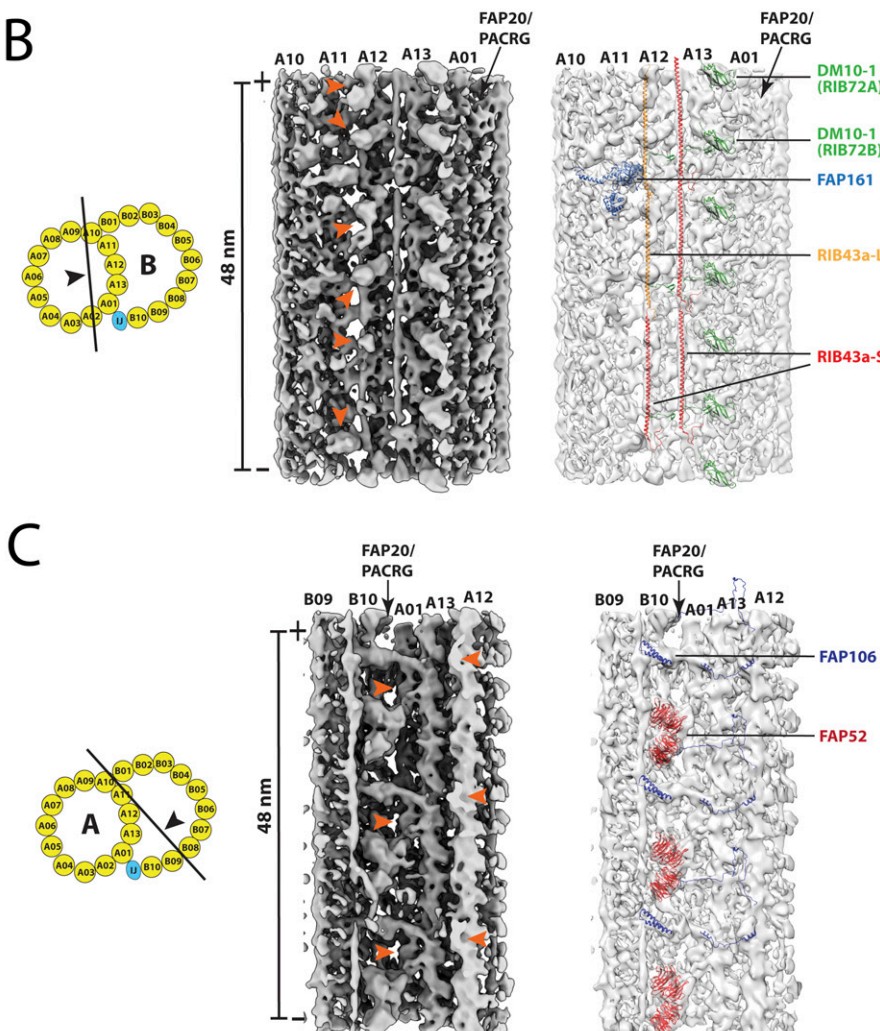

**Figure 5. Structure comparison of the doublet microtubule between *Tetrahymena* and *Chlamydomonas*.**
Three regions in the DMT are compared. These are as follows: **(A)** the A-tubule "seam" region spanning the pf A06~A10, **(B)** the ribbon region, and **(C)** the inner junction region. In each panel, *left*: a cartoon depicts the viewing direction of the structures in the DMT; *middle*: the red arrowheads indicate the unidentified microtubule inner protein densities that are unique to the *Tetrahymena* DMT; *right*: the microtubule inner proteins found in both organisms are indicated. The atomic models from *Chlamydomonas* are fit into the *Tetrahymena* density maps in light gray.

The shapes and locations are consistent with the structure in *Chlamydomonas*. Last, we found a likely density for FAP161, a MIP binding at the A-tubule seam. However, its EF-hand motif had to be adjusted to better fit into our map (Fig 5A). Surprisingly, there were at least six globular densities in our map that we could not find their corresponding models in *Chlamydomonas* DMT. Five were in the seam region, including two globular densities previously assigned as MIP2b~c (Song et al, 2020). Another density laterally extended

from pf A05 to A07 (Fig 5A). Conversely, we could not fit eight MIP models from *Chlamydomonas* into our map. These MIPs are: FAP182, FAP129, FAP85, FAP21, FAP68, FAP143, FAP95, and FAP141 (Fig S4A). Likely, this is in part due to the limited resolution of our map and the unstructured nature of the extended MIPs, such as FAP129 and FAP143.

Next, we examined the pf A10~A01 region, including the ribbon region shared by the A- and B-tubule. Recent single particle cryoEM studies have identified two filamentous structures in the ribbon region, running longitudinally on the cleft between pf A11/A12 and A12/A13, respectively (Ichikawa et al, 2019; Ma et al, 2019). Each filament is composed of multiple copies of RIB43a connected in a head-to-tail fashion. In *Chlamydomonas*, the filaments in both positions were composed of RIB43a. However, *Tetrahymena* has two isoforms, RIB43a-S and RIB43a-L. Whereas the filament at A12/A13 was composed solely of RIB43a-S, the filament at A111/A12 was composed of alternation of RIB43a-S and RIB43a-L (Ichikawa et al, 2019). We confirmed this by fitting both structures in our map (Fig 5B). Besides RIB43a-S/L, the two organisms displayed marked differences in this region. We could not find models for six globular densities that were associated with RIB43a-L/RIB43a-S filament (Fig 5B). Five of them have been assigned previously as MIP4a~e (Song et al, 2020). Meanwhile, five MIPs from *Chlamydomonas*, FAP166, FAP273, FAP363, RIB21, and RIB30, could not find corresponding

density in our structure (Fig S4B). Given the difference in RIB43a between these two organisms and that many MIPs were directly associated with the RIB43a, the observed structural differences are not completely surprising.

Last, we examined the inner junction region. Previous studies showed structural differences between *Tetrahymena* and *Chlamydomonas* in this region (Ma et al, 2019; Khalifa et al, 2020). Our modeling study confirmed this. We identified FAP52 and FAP106 in our map (Fig 5C). However, our structure does not show densities corresponding to FAP126 and FAP276. Conversely, we found a second "tether density" linking FAP52 to pfA13 and a 16-nm periodic structure on the ridge of pf A12 facing the lumen of B-tubule (Fig 5C). Both are unique to *Tetrahymena*. Previously the MIP on pf A12 was assigned as MIP5 (Song et al, 2020).

Surprisingly, we uncovered an MIP at the *Tetrahymena* inner junction that was not found in *Chlamydomonas*. Previously, this MIP was indicated as a minor density (Ichikawa et al, 2017; Song et al, 2020). However, by multiple iterations of classification and alignment, we show this MIP is in fact a large multi-domain protein or complex, with an estimated molecular weight of ~130 KDa. It has an overall X shape with 48-nm longitudinal periodicity. It spans multiple pfs across the inner junction, from B07 to A13. Longitudinally, it is ~16 nm wide (Figs 6A and S5 and Video 5). To further characterize the

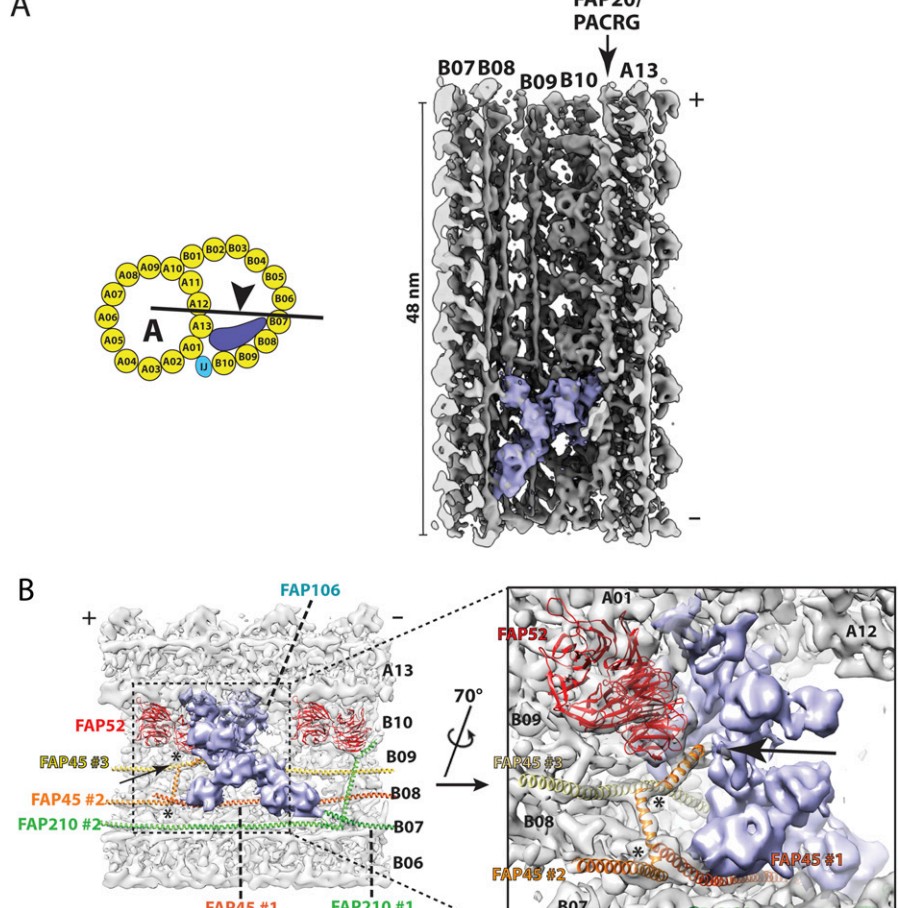

**Figure 6. An unidentified microtubule inner protein (MIP) at the inner junction of the doublet microtubule.**
**(A)** An unidentified MIP (royal blue) in the inner junction region of DMT. In the cartoon, an arrowhead indicates the direction of view of the MIP. This multi-domain MIP spans several protofilaments, from B07 to A13, longitudinally repeating every 48 nm. **(B)** An unidentified MIP (royal blue) makes extensive interactions with multiple protofilaments and other MIPs at the inner junction. On the left, the plus end of DMT is towards the left as indicated. Two FAP52 (red) are shown. A FAP106 is behind the unidentified MIP (not shown). Three FAP45 are shown. FAP45 #1 (dark orange) and FAP45 #3 (yellow) bind on the cleft between protofilament B07/B08 and B08/B09, respectively. The α-helical FAP45 #2 (orange) initiates from the pf B09, threads through a gap between FAP52 and the unidentified MIP (indicated by an arrow), makes two ~90° turns (indicated by two asterisks). FAP45 #2 interacts with FAP45 #3 and FAP45 #1 before landing on the cleft between pf B07/B08. Two FAP210 are shown in green. The inset shows a magnified view of the described interactions among the unidentified MIP (royal blue), FAP45 #2 and other nearby MIPs.

interaction of this MIP with other components in the inner junction, we fit models of FAP52, FAP45, and FAP210, into our density map (Fig 6B and Video 6). This MIP contacts the models at multiple points. Specifically, it binds to the FAP210 at pf B06/B07, it also binds to two copies of FAP45 on the clefts between pf B07/B08 and B08/B09 (Fig 6B and Video 6). In addition, the MIP interacts with FAP106 and pf A13. Interestingly, the MIP contacts FAP52 via FAP45. The N-terminal helix of FAP45 threads through a narrow gap between this MIP and the FAP52, making contacts with both. This FAP45 has a kink at this point; it continues and makes a ~90-degree bend into the transverse direction where it interacts with a second molecule of FAP45. When the helix of FAP45 in the transverse direction reaches the inner cleft of pf B07/B08, it takes a second 90-degree sharp bend, switching back to the longitudinal direction towards the plus end. At this point, it contacts the C-terminal end of a third FAP45 molecule (Fig 6B), forming a longitudinal head-to-tail connection, a common theme shared by many fMIPs in the axoneme (Ichikawa et al, 2019; Ma et al, 2019). To find out if this MIP had any preferential binding to any particular DMT, we mapped the structure on the axonemes. We could not detect any radial asymmetry or any DMT preference, instead, this MIP was found in all nine DMTs in the axoneme. Taken together, we have identified a large MIP complex at the inner junction that is unique to the *Tetrahymena* DMT. It makes extensive interactions with multiple MIPs and the luminal wall of DMT. In the future, it will be interesting to identify the molecular nature of the MIP and to understand its function.

## Discussion

### FAP115 functions as a "molecular linker" essential for structural integrity of the DMT

The EF-hand protein FAP115 was identified as one of the MIPs lost in *RIB72A/B* knockout mutants (Stoddard et al, 2018; Ma et al, 2019; Fabritius et al, 2021). In this study, we provide a detailed description of FAP115 binding to the DMT: The protein bridges pf A02 and A03 on the luminal wall of the A-tubule, making multiple interactions with the adjacent proteins; its NTD interacts with $\alpha/\beta$ tubulin in pf A02. The CTD anchors at the corner of four-tubulin dimers, stabilizing both lateral and longitudinal MT interfaces at pf A02 and A03. In addition, FAP115 interacts with either RIB72A or RIB72B via an extended loop. We further analyzed the axonemes isolated from a *FAP115KO* mutant. Strikingly, compared to the wild-type, 40% of DMT showed structural defects beyond the FAP115 binding site. The defects were identified in the pf A04~A05 region, the ribbon region and the inner junction region. Moreover, these defects propagated forming long stretches of defective DMT in the axoneme, and likely causing the motility defect observed in the mutant (Fabritius et al, 2021). Interestingly, despite extensive interactions between FAP115 and the luminal wall of DMT, FAP115 was absent in the *rib72* null mutations (Stoddard et al, 2018; Ma et al, 2019), suggesting RIB72A/B are essential for recruitment of FAP115 to the axoneme and their binding might take place before FAP115 assembly on the axoneme. Conversely, we did not observe any significant reduction in RIB72A/B in the *FAP115KO* mutant. Taken together, these findings, consistent with previous studies on the other part of axoneme, suggest that the axoneme assembly is a sequential and tightly controlled process (Qin et al, 2004; Diener et al, 2011).

Perhaps, one of the most striking and interesting findings in the mutant is the extensive defects observed beyond the FAP115 binding site in the DMT and the propagation of the defects along the axoneme. This is likely a secondary effect caused by the FAP115 depletion. A similar effect has been observed previously with depletion of other MIPs. For example, in *Chlamydomonas* it was found that loss of inner junction components PACRG and FAP20 caused reduction in the beak and the IDA structure that were distant from the inner junction (Yanagisawa et al, 2014; Dymek et al, 2019). Currently, we lack knowledge on the timing of when these defects occurred in the axoneme. It could take place during cilia assembly in the *FAP115KO* mutant where the absence of FAP115 might affect recruitment of other MIPs. More likely the defects were introduced during cilia beating when the axoneme experienced substantial mechanical stress. Because the MIPs bind to DMT nonuniformly and the axoneme DMT are in a radially asymmetric shape, the curvatures between laterally associated pfs vary and might be finely tuned by the associated MIPs (Ichikawa et al, 2017, 2019). Notably, besides the seam region, the pf A02/A03, where FAP115 binds, has the highest local inter-protofilament curvature in the DMT, a specific geometry that fits FAP115 well. During cilia beating, a disruption of the local curvature or geometry due to the depletion of MIP(s), such as FAP115, could have a profound and long-range effect on the DMT structure. This is consistent with previous in vitro experiments on MT, where local defects can have a long-range effect on distant sites (Schaedel et al, 2015; Rai et al, 2021 *Preprint*).

Microtubules and axonemes are intrinsically stiff polymers (Sale & Satir, 1976; Schaedel et al, 2015). However, local bending and flexibility are necessary for generating waveform. Therefore, there exists a fine balance between the microtubule's intrinsic stiffness and the plasticity necessary for local bending and sliding during cilia beating. It is likely that one of the roles for many MIPs, such as FAP115, is to fine-tune the physical properties of MT, and to insulate any defect introduced under mechanical stress, preventing the defects from propagating. Overall, we propose that FAP115 functions as a "molecular linker" strengthening the local DMT lattice by bolstering inter- and intra-protofilament interfaces, making DMT more resilient to defects introduced during cilia beating.

### Structural conservation and diversity of MIPs in two model organisms

*Tetrahymena* and *Chlamydomonas* represent two phylogenetically distant groups, ciliates and green algae, which were estimated to have diverged more than a billion years ago (Kumar et al, 2017).

Remarkably, the axoneme of the motile cilium/flagellum is one of the most conserved structures during the evolution of the eukaryotic branches of the phylogenetic tree of life (Cavalier-Smith, 2002; Satir & Christensen, 2007; Satir et al, 2008; Carvalho-Santos et al, 2011; Mitchell, 2016; Bornens, 2018). In addition to comparative genomic and proteomic analyses in different organisms (Ostrowski et al, 2002; Avidor-Reiss et al, 2004; Li et al, 2004; Pazour et al, 2005; Smith et al, 2005; Fabritius et al, 2021), structural comparisons of cilia among different organisms have been carried out previously,

showing both evolutionary conservation and diversity of this organelle (Nicastro et al, 2011; Pigino et al, 2012; Carbajal-González et al, 2013; Lin et al, 2014; Khalifa et al, 2020). However, most studies were limited to morphological descriptions, partly because of the limited resolution in the structures. Here, in the light of the improved resolution in the axoneme structures, we take an initial step to systematically examine the presence of MIPs in axonemes from *Chlamydomonas* and *Tetrahymena*. Our goal is to reveal structural conservation and diversity of the MIPs and to have a deeper understanding of their functions.

The results are summarized in Table 2. In the study, we found 19 MIPs that were shared by both organisms. Interestingly, among these 19 MIPs, except FAP112, FAP115, and DC3, 16 have found their human orthologue proteins, 13 are ciliopathic in humans or cause phenotypic defects in other metazoan organisms. This indicates that these MIPs are evolutionarily conserved and have critical functions in motile cilia in many organisms or species. For example, FAP53, FAP127, and FAP161 are near the seam of the A-tubule (pf A09~A10), which might provide critical structural roles. PACRG, FAP20, FAP52, and FAP106 are components of the inner junction and essential for its structural stability (Yanagisawa et al, 2014; Dymek et al, 2019; Ma et al, 2019; Owa et al, 2019; Khalifa et al, 2020). FAP67, a nucleoside diphosphate kinase that binds to the DMT via its DM10 domain, has been postulated to have a role in maintaining the local GTP concentration that is essential for axonemal homeostasis (Ma et al, 2019). RIB72A/B and RIB43a-S/L, which span several protofilaments laterally or form head-to-tail longitudinal filaments, might have critical roles in structural stability of the DMT as well as providing binding scaffolds for other MIPs (Norrander et al, 2000; Stoddard et al, 2018). CCDC39/FAP59 and CCDC40/FAP172 form a molecular ruler, essential for establishing and maintaining the 96-nm periodicity of the axoneme (Oda et al, 2014). Given the structural conservation of these MIPs that we observed and the phylogenic distance between *Tetrahymena* and *Chlamydomonas*, these are likely to be a set of ancestral MIPs that had been present in the last eukaryotic common ancestor (LECA) (Mitchell, 2016).

Meanwhile, we also found 19 MIPs or densities that are unique to either *Tetrahymena* or *Chlamydomonas*, showing marked differences in DMT structures between these two organisms. Consistent with the structure differences, among the MIPs identified

in *Chlamydomonas* but missing in our structure from *Tetrahymena*, many have their corresponding genes absent in *Tetrahymena* genome database. These genes are *FAP182*, *FAP129*, *FAP85*, *FAP21*, *FAP68*, *FAP143*, *FAP95*, *FAP166*, *FAP273*, and *FAP363*. Their absence in both structure and genome indicates that these set of MIPs are diverged during evolution.

The morphological differences of axonemes among different evolutionary groups have been noticed previously. It has been implied that the structural differences might be linked to the variation of waveform (Pigino et al, 2012; Kirima & Oiwa, 2017). Interestingly, the flagella in *Tetrahymena* adapted an asymmetric planar whip-like waveform, whereas *Chlamydomonas* has both a planar whip-like waveform in forward locomotion and a sinusoidal-like symmetric waveform when swimming backwards. It will be intriguing to speculate if any subset of these MIPs might attribute to a particular type of waveform.

Taken together, by comparing the DMT structures from two model organisms, *Tetrahymena* and *Chlamydomonas*, we have identified a group of MIPs that are present in both structures as well as subsets that are unique to each organism, showing their structural conservation and diversity. In the future it will be interesting to expand this comparative structure study to other organisms and to gain insight into the function of this ancient and remarkable organelle in the course of evolution.

# Materials and Methods

### Isolation of axoneme from the *Tetrahymena* strains

The wild-type (B2086.2) *Tetrahymena* strain was obtained from the Tetrahymena Stock Center (Cornell University). All strains were grown in the standard medium 2% SPP (2% protease peptone, 0.1% yeast extract, 0.2% glucose, 0.003% FeCl$_3$) at 30°C. The *FAP115* knockout strain was generated as described previously (Fabritius et al, 2021).

To isolate axoneme from the cells, cells grown in 500 ml 2% SPP media were harvested by spinning down at 1,100*g* for 5 min in room temperature. Cells were washed with Tris–HCl (pH 7.5) and span down again at 1,100*g* for 5 min. Cells were suspended in pH-Shock

**Table 2. Comparison of microtubule inner protein (MIP) structure between *Tetrahymena t.* and *Chlamydomonas r.***

| | |
|---|---|
| 19 MIPs found in both *Tetrahymena t.* and *Chlamydomonas r.* | RIB72, FAP115, FAP67, FAP53, FAP127, FAP161, RIB43, FAP20 (fly), PACRG, FAP52, FAP106, FAP45, FAP210, FAP112, CCDC39/FAP59, CCDC40/FAP172, DC1, DC2, DC3 |
| 19 MIPs identified in *Chlamydomonas r.* but possibly absent in *Tetrahymena t.* | FAP222, FAP252, FAP21, FAP68, FAP107, FAP85, FAP90, FAP95, FAP129, FAP141, FAP143, FAP182 (mice), FAP166, FAP273, FAP363, RIB21, RIB30, FAP126 (mice), FAP276 |
| 19 Unidentified MIPs or complexes found in *Tetrahymena t.* whose density are missing in *Chlamydomonas r.* and their atomic models are lacking, | 1 in pf A04~A05 region<br>6 in pf A06~A10 region<br>6 in *ribbon region*<br>3 in inner junction region<br>3 fMIPs in pf B03~B06 |

Note: The MIPs highlighted in red cause ciliopathy in human or phenotypic defects in other metazoan as indicated. These are based on a summary in Ma et al (2019) and references cited within.

Buffer (10 mM Tris, pH 7.5, 50 mM sucrose, 10 mM CaCl$_2$, and 0.33% Proteinase Inhibitor Cocktail (Sigma-Aldrich)). 0.5 M acetic acid was added to lower the pH to 4.3. After one minute, the pH was brought back to 7.5 by adding 1 M KOH (Sigma-Aldrich). The deflagellated cells were spun at 1,500$g$ for 5 min, in 4°C. The supernatant was centrifuged twice at 1,860$g$ for 5 min in 4°C. The cilia in the supernatant were pelleted by spinning at 10,000$g$ for 15 min. The cilia pellet was resuspended in cold HMEEK buffer (30 mM Hepes, 25 mM KCl, 5 mM MgSO$_4$, 0.1 mM EDTA, and 1 mM K-EGTA). 1% Igepal CA 630 (Sigma-Aldrich) was added to demembrane the axoneme. After rotating for 20 min in 4°C, the sample was centrifuged at 10,000$g$ for 10 min. The pellet containing the axoneme was carefully resuspended in cold HMEEK buffer.

### Electron cryo-tomography data collection

To make grids for cryoET, the isolated ciliary axonemes were mixed with 10 nm colloid gold (Sigma-Aldrich). 4 $\mu$l sample was applied onto Quantifoil grid Cu/Rh 200 R2/2 (Quantifoil, Inc) and was flash-frozen into liquid ethane using a Vitrobot (FEI, Inc). Tomography tilt series were collected on a field emission gun microscope (Titan Krios, FEI, Inc) running at 300 kV. A Bio-Quantum GIF post-column energy filter (Gatan, Inc) was used during data collection. The slit width was set at 20 eV. SerialEM (Mastronarde, 2005) was used for collecting tomography tilt series at a nominal magnification of 33,000. Images were recorded on a K3 direct electron detector (Gatan, Inc) in super-resolution and dose-fractionation mode. The effective physical pixel size on image was 2.65 Å. The dose rate was set at 20 electron/pixel/second. The specimen was tilted in a bi-directional scheme, starting from zero degree, first tilted towards –60°, followed by a second half from +2° to +60°, in 2° increment per step. The total accumulative dose on the sample was limited to 80 electron/Å$^2$.

### Data processing for subtomogram averaging and model building

For tomogram reconstruction and subtomogram averaging, the dose-fractionated movie at each tilt in the tilt series was corrected of motion and summed using MotionCor2 (Zheng et al, 2017). The tilt series were aligned based on the gold beads as fiducials by using IMOD (Kremer et al, 1996) and TomoAlign (Fernandez et al, 2018). The contrast transfer function for each tilt series was determined and corrected by TomoCTF (Fernández et al, 2006). The tomograms were reconstructed by TomoRec (Fernandez et al, 2019) taking into account of the beam-induced sample motion during data collection. Total 51 wild-type axoneme tomograms from 49 tilt series and 65 *FAP115* KO axoneme tomograms from 64 tilt series were selected for reconstruction and subtomogram averaging.

To identify the DMT for subtomogram averaging, the 6xbinned tomograms were used. The center of DMT and the approximate orientation of the axoneme relative to the tilt axis were manually annotated in a Spider metadata file (Frank et al, 1996). The initial subtomogram alignment and average were carried out in 2xbinned format (pixel size 5.30 Å). The longitudinal segment length of axoneme DMT in subtomogram was limited to 24 nm and 50% overlapping with neighboring segments. Without using any external reference, the subtomogram alignment was carried out by a program MLTOMO implemented in the Xmipp software package (Scheres et al, 2009).

Because DMT from axoneme was a continuous filament up to several microns in length, after obtaining initial alignment parameters, a homemade program RANSAC was used to detect and remove any alignment outliers and to impose the continuity constraint on the neighboring segments. It corrected the mis-aligned subtomograms by regression. MLTOMO and Relion (Bharat & Scheres, 2016) were extensively used for focused classification of the subtomograms. This was critical for determining the correct periodicity of the MIPs and for identifying structure defects or heterogeneity in the DMT. In the focused classification, a particular MIP structure in a subregion of DMT was chosen for discerning different classes. Specifically, in the pf A01~A06 region, the un-identified MIP with 16 nm periodicity that crosslinked RIB72A/B was used as a fiducial to identify the out-of-register classes; in the A-tubule seam region, two copies of FAP67 were used as fiducials for identifying the local 48 nm periodicity; in the inner junction region, the unidentified MIP spanning multiple pfs with 48 nm periodicity was used as a fiducial. The focused classification results were further confirmed by positions of other MIPs in the region. Because the assembly of DMT is a coherent process, the relative position of a particular MIP to other MIPs will be consistent. Once a MIP's periodicity was determined, the out-of-register subtomograms were re-centered and re-extracted. This was followed by combining all subtomograms for the next round of refinement.

For final refinement and averaging, the subtomograms were re-extracted from tomogram volumes in unbinned format (pixel size 2.65 Å). During the extraction, a customized, soft-edged, and cy-lindrical shaped binary mask, centered on a particular region or the structure of interest in the DMT, was imposed onto the extracted subtomograms. This was carried out by the program Spider (Frank et al, 1996). Except the refinement of the unidentified MIP in the inner junction where the DMT longitudinal length was arbitrarily set at 24 nm, in all other cases, the extracted DMT segment length was set to be 18 or 53 nm, containing a full repeat of the structure that was in either 16 or 48 nm periodicity as determined by the focused classification. The re-extracted and masked subtomo-grams, containing only the structure of interest, for example, the pf A01~A06 region in the A-tubule or the inner junction region, were aligned in a two-independent-datasets scheme implemented in Relion (Scheres & Chen, 2012). The refinement took into account dose-weighting and thickness weighting (Bharat & Scheres, 2016). The overall resolutions were reported based on the Fourier Shell Correlation cutoff at 0.143 (Rosenthal & Henderson, 2003). The local resolution assessments were carried out by the program *blocres* in Bsoft (Heymann & Belnap, 2007). This is particularly useful for assessing the resolution anisotropy of large structures such as the DMT.

A composite map is useful for putting together individual component maps, often in higher resolution, in the context of large assemblage. The composite maps were generated in UCSF Chi-mera (Pettersen et al, 2004). First, the component maps were aligned based on their mutual overlapping regions by using UCSF Chimera's "fit-in-map" function. This was followed by taking maximum density values from the overlapping voxels and com-bining the component maps into a composite map. UCSF Chimera

and ChimeraX (Pettersen et al, 2021) were used for visualization, model building and for recording images and videos.

The pseudo-atomic models for different parts of the DMT were built in Chimera or PyMol by fitting previously published atomic models into the subtomogram averaging density maps.

16 structures, including their Electron Microscopy Data Bank (EMDB) access codes are summarized in Table 1.

## Data Availability

The EM structures have been deposited in the EMDB with the following accession numbers: EMD-24364, EMD-24366, EMD-24367, EMD-24368, EMD-24370, EMD-24371, EMD-24372, EMD-24373, EMD-24374, EMD-24375, EMD-24376, EMD-24377, EMD-24379, EMD-24380, EMD-24381, and EMD-24382.

## Supplementary Information

## Acknowledgements

We thank Zanlin Yu and David Bulkley (UCSF) for assistance on tomography data collection, Wynton HPC team at UCSF for supporting the computational infrastructure, Daniela Nicastro (UT, Southwestern) for kindly providing a *Tetrahymena* axoneme purification protocol, and Tom Goddart (UCSF) for help with UCSF Chimera software. We also thank Fei Guo (UC, Davis) for help preparing cryo-grids. We are grateful to many of our colleagues for critical reading of the manuscript and for their encouragements. This work is supported in part by National Institutes of Health (NIH) grants R01GM127571 (M Winey), R35GM118099 (DA Agard), and by the Spanish AEI/FEDER (SAF2017-84565-R) (J-J Fernandez).

### Author Contributions

S Li: data curation, formal analysis, methodology, and writing—original draft.
J-J Fernandez: software, formal analysis, funding acquisition, validation, investigation, methodology, and writing—review and editing.
AS Fabritius: data curation and writing—review and editing.
DA Agard: resources, supervision, and funding acquisition.
M Winey: conceptualization, resources, supervision, funding acquisition, and writing—review and editing.

### Conflict of Interest Statement

The authors declare that they have no conflict of interest.

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
