## [Reviewer comments · Life Science Alliance]

Life Science Alliance

Electron Cryo-Tomography Structure of Axonemal Doublet Microtubule from *Tetrahymena thermophila*

Sam Li, José J. Fernández, Amy Fabritius, David Agard, and Mark Winey

DOI: <https://doi.org/10.26508/lsa.202101225>

Corresponding author(s): Sam Li, University of California, San Francisco

Review Timeline:

Submission Date:	2021-09-02
Editorial Decision:	2021-10-13
Revision Received:	2021-11-04
Editorial Decision:	2021-11-26
Revision Received:	2021-12-03
Accepted:	2021-12-06

Scientific Editor: Novella Guidi

Transaction Report:

October 13, 2021

Re: Life Science Alliance manuscript #LSA-2021-01225-T

Dr. Sam Li
San Francisco, Univ. of California at
HHMI, Dept. of Biochemistry
Univ. California San Francisco
600, 16th Street
San Francisco, California 94158

Dear Dr. Li,

Thank you for submitting your manuscript entitled "Electron Cryo-Tomography Structure of Axonemal Doublet Microtubule from *Tetrahymena thermophila*" to Life Science Alliance. The manuscript was assessed by expert reviewers, whose comments are appended to this letter. As you will note from the reviewers' comments below, both reviewers are quite positive about the study and feels that the paper is technically sound and a good advancement of the structural study in the cilia/flagella field. They both requests few minor revisions of text and figures. We, thus, encourage you to submit a revised version of the manuscript back to LSA that responds to all of the reviewers' points.

Thank you for this interesting contribution to Life Science Alliance. We are looking forward to receiving your revised manuscript.

Sincerely,

-- By submitting a revision, you attest that you are aware of our payment policies found here: <https://www.life-science->

B. MANUSCRIPT ORGANIZATION AND FORMATTING:

Reviewer #1 (Comments to the Authors (Required)):

Microtubule inner proteins (MIPs) of cilia were identified first in 2006. Recent advances has allowed single particle cryo-EM doublet microtubules to achieve higher resolution and identification of many MIP in the green alga, *Chlamydomonas*, and in the ciliate, *Tetrahymena*.

This manuscript uses cryo electron tomography to examine MIPs in two mutant strains in *Tetrahymena* and to compare structures in *Tetrahymena* to MIPs from *Chlamydomonas*.

They confirm results from other labs that the MIPs are highly interconnected and a loss of function mutant in Fap115 causes extensive damage but not complete. They find conservation between *Tetrahymena* and *Chlamydomonas* as well as differences.

Comments

The result that the MIPs are different between *Tetrahymena* and *Chlamydomonas* is interesting but they authors should provide information that the genes for these MIPs are absent. This should include FAP182, FAP129, FAP85, FAP21, FAP68, FAP143, FAP95, FAP166, FAP273, and FAP363.

For example, FAP182 is a very divergent protein but the structures (by AlphaFold) from human to *Chlamydomonas* are very similar. Another interesting example is the inner junction proteins of the basal body body. WDR90 appears to be a "fusion" of two different proteins in the ciliary axoneme. For their large unknown complex, did they examine this by similarity of sequence/structure?

The other unique finding is the phenotype of the knockout of FAP115. The heterogeneity of the phenotype is interesting and should be mentioned in the summary as it is one of the novel findings.

Minor comments

Page 8. The authors say that K40 is a potential site for post-translational modification. Do they think that *Tetrahymena* has a different lysine that is modified? LeDizet and Piperno in 1987 sequenced alpha tubulin and showed that K40 in *Chlamydomonas* is acetylated.

The nomenclature used is not consistent. Most *Tetrahymena* proteins are Fap115, but there are exception. PACRG is all upper case on page 6 and RIB43a-S and RIB43a-L is all upper case on page 16 and then lower case.

Genes should be italics. On page 9, the RIB72 null mutations should be *rib72* null mutations. At least it should be italics and maybe not lower case Not being able to find nomenclature rules for *Tetrahymena*, are mutant alleles lower case? On the same the FAP115 knockout mutation is upper case. Page 11. FAP115 knockout should be in italics as it is the gene.

When the human protein EFHC1 is mentioned, the correct naming is used. The authors should use the correct naming for *Chlamydomonas* genes. All proteins are upper case and Roman.

The term molecular staple was used in Ma et al., 2019 for different proteins. I think the use is confusing for the field to have other proteins called molecular staples.

The Nicastro lab showed convincingly that the dynein regulatory complex is part of the nexin link and thus the name has become the N-DRC and not just the DRC.

The spectrum of diseases of motile cilia should include infertility. (page 3)

The term partition breaks with tradition. Dick Linck who studied the shared wall called it the ribbon (Ikeda et al., 2003). Ribbon

should be mentioned.

Page 6: I think there is a word missing/misused in this sentence. ... we initially focused on the DMT with a 16 nm repeat. I think they mean the MIPs with a 16 nm repeat.

Reviewer #2 (Comments to the Authors (Required)):

This paper describes the ~10 Å resolution structures of the doublet microtubule of *Tetrahymena* cilia (abbreviated as T-DMT), which were obtained by subtomographic averaging of cryo-electron tomographs. A wild type structure of T-DMT was resolved at 4-5 Å resolution using cryo-EM and single particle analysis (Ichikawa et al., 2017, 2019). Because T-DMT by Ichikawa et al are treated with sonication and/or sarkosyl, some of the components can be lost during the disintegration of axoneme into DMTs. Therefore, it can be closer to the native structure, although the resolution of T-DMT in this paper is lower than previous ones. For example, the authors find a new unidentified ~130 kDa MIP (microtubule inner protein) at the inner junction, as shown in Figure 6. This new MIP is interesting because it connects between the inner junction and the further part of B-tubules. In addition, the authors analyzed the structure of FAP115KO mutant DMT. The phenotype of this FAP115KO was recently analyzed and published by the authors' group. Therefore the structural analysis is a good complement for the analysis. Overall, the paper is technically sound and a good advancement of the structural study in the cilia/flagella field. The reviewer recommends this paper for publication, with some minor revisions of texts and figures.

Major points:

(1) Please explain or discuss why the new ~130 kDa MIP at the inner junction was not found (or indicated as minor density) in the previous publications.

What percentage of the DMT were used to show the new MIP?

(2) Please show the difference between their T-DMT and the ones in the previous publication (probably in the supplementary figure) and discuss their technical advantages and/or disadvantages.

Minor points:

(1) Figure 4G does not clearly show the colors of dots, because they are too small. In addition, the near and far side dots of DMT are overlapping. Please consider a better presentation, such as an unfolded form of DMT.

(2) Table 1: "Number of subtomograms" should be "Number of subtomographic particles"

Ichikawa, M. et al. (2017) 'Subnanometre-resolution structure of the doublet microtubule reveals new classes of microtubule-associated proteins', *Nature Communications*, 8, p. 15035. doi:10.1038/ncomms15035.

Ichikawa, M. et al. (2019) 'Tubulin lattice in cilia is in a stressed form regulated by microtubule inner proteins', *Proceedings of the National Academy of Sciences of the United States of America*, 116(40), pp. 19930-19938. doi:10.1073/pnas.1911119116.

Ma, M. et al. (2019) 'Structure of the Decorated Ciliary Doublet Microtubule', *Cell*, 179(4), pp. 909-922.e12.

doi:10.1016/j.cell.2019.09.030.

Re: Life Science Alliance manuscript #LSA-2021-01225-T

Thank you for submitting your manuscript entitled "Electron Cryo-Tomography Structure of Axonemal Doublet Microtubule from Tetrahymena thermophila" to Life Science Alliance. The manuscript was assessed by expert reviewers, whose comments are appended to this letter. As you will note from the reviewers' comments below, both reviewers are quite positive about the study and feels that the paper is technically sound and a good advancement of the structural study in the cilia/flagella field. They both requests few minor revisions of text and figures. We, thus, encourage you to submit a revised version of the manuscript back to LSA that responds to all of the reviewers' points.

We thank both reviewers for their positive response and their constructive comments. Below, we address their comments and concerns, point-by-point.

Reviewer #1 (Comments to the Authors (Required)):

Microtubule inner proteins (MIPs) of cilia were identified first in 2006. Recent advances has allowed single particle cryo-EM doublet microtubules to achieve higher resolution and identification of many MIP in the green alga, Chlamydomonas, and in the ciliate, Tetrahymena. This manuscript uses cryo electron tomography to examine MIPs in two mutant strains in Tetrahymena and to compare structures in Tetrahymena to MIPs from Chlamydomonas.

They confirm results from other labs that the MIPs are highly interconnected and a loss of function mutant in Fap115 causes extensive damage but not complete. They find conservation between Tetrahymena and Chlamydomonas as well as differences.

Comments

The result that the MIPs are different between Tetrahymena and Chlamydomonas is interesting but they authors should provide information that the genes for these MIPs are absent. This should include FAP182, FAP129, FAP85, FAP21, FAP68, FAP143, FAP95, FAP166, FAP273, and FAP363.

For example, FAP182 is a very divergent protein but the structures (by AlphaFold) from human to Chlamydomonas are very similar. Another interesting example is the inner junction proteins of the basal body body. WDR90 appears to be a "fusion" of two different proteins in the ciliary axoneme. For their large unknown complex, did they examine this by similarity of sequence/structure?

Response:

We thank reviewer #1 for the insight! For 19 *Chlamydomonas* MIPs that were found absent in our *Tetrahymena* structure (listed in Table 2), our BLAST searches were not able to identify their homolog genes or protein in *Tetrahymena*. The result indicates these group of MIPs are indeed absent or very diverged, consistent with the observation in our structure comparison.

Now we have added this information in the discussion section in our revised manuscript, where we state –

“Consistent with the structure differences, among the MIPs identified in *Chlamydomonas* but missing in our structure from *Tetrahymena*, many have their corresponding genes absent in *Tetrahymena* genome database. These genes are *FAP182*, *FAP129*, *FAP85*, *FAP21*, *FAP68*, *FAP143*, *FAP95*, *FAP166*, *FAP273*, and *FAP363*. Their absence in both structure and genome indicates that these set of MIPs are diverged during the evolution.”

The other unique finding is the phenotype of the knockout of FAP115. The heterogeneity of the phenotype is interesting and should be mentioned in the summary as it is one of the novel findings.

Response:

We have now added a sentence in the abstract –

“We observed substantial structure heterogeneity in DMT in a *FAP115* knockout strain, showing extensive structural defects beyond the *FAP115* binding site.”

Minor comments

Page 8. The authors say that K40 is a potential site for post-translational modification. Do they think that Tetrahymena has a different lysine that is modified? LeDizet and Piperno in 1987 sequenced alpha tubulin and showed that K40 in Chlamydomonas is acetylated.

Response:

We believe the α K40 is acetylated in both *Tetrahymena* and *Chlamydomonas*. To avoid confusion, we have modified our statement as –

“Interestingly, the α K40 residue is a site for post-translation modification by acetylation (Akella et al., 2010).”

The nomenclature used is not consistent. Most Tetrahymena proteins are Fap115, but there are exception. PACRG is all upper case on page 6 and RIB43a-S and RIB43a-L is all upper case on page 16 and then lower case.

Genes should be italics. On page 9, the RIB72 null mutations should be rib72 null mutations. At least it should be italics and maybe not lower case Not being able to find nomenclature rules for Tetrahymena, are mutant alleles lower case? On the same the FAP115 knockout mutation is upper case. Page 11. FAP115 knockout should be in italics as it is the gene.

When the human protein EFHC1 is mentioned, the correct naming is used. The authors should use the correct naming for Chlamydomonas genes. All proteins are upper case and Roman.

Response:

We thank Reviewer #1 for pointing this out. We have changed our gene and protein convention accordingly. All *Tetrahymena* MIP gene names are in italic, wildtypes are in upper case and mutants are in lower case.

All MIP names are now in upper case and Roman, such as RIB72A.

The term molecular staple was used in Ma et al., 2019 for different proteins. I think the use is confusing for the field to have other proteins called molecular staples.

The Nicaastro lab showed convincingly that the dynein regulatory complex is part of the nexin link and thus the name has become the N-DRC and not just the DRC.

Response:

We agree with the reviewer. To avoid confusion, instead of calling it “a molecular staple” we now describe FAP115 as a “molecular linker”, which is a more proper description of its function, contacting and crosslinking multiple proteins in the lumen of A-tubule.

The spectrum of diseases of motile cilia should include infertility. (page 3)

Response:

We have included infertility as one of the human diseases caused by defective motile cilia.

The term partition breaks with tradition. Dick Linck who studied the shared wall called it the ribbon (Ikeda et al., 2003). Ribbon should be mentioned.

We now call this region ribbon

Page 6: I think there is a word missing/misused in this sentence. ... we initially focused on the DMT with a 16 nm repeat. I think they mean the MIPs with a 16 nm repeat.

Response:

We thank Reviewer #1 to point this out. Now we have changed it to “We initially focused on the MIPs with 16 nm periodicity”

Reviewer #2 (Comments to the Authors (Required)):

This paper describes the ~10 Å resolution structures of the doublet microtubule of *Tetrahymena* cilia (abbreviated as T-DMT), which were obtained by subtomographic averaging of cryo-electron tomographs. A wild type structure of T-DMT was resolved at 4-5 Å resolution using cryo-EM and single particle analysis (Ichikawa et al., 2017, 2019). Because T-DMT by Ichikawa et al are treated with sonication and/or sarkosyl, some of the components can be lost during the disintegration of axoneme into DMTs. Therefore, it can be closer to the native structure, although the resolution of T-DMT in this paper is lower than previous ones.

For example, the authors find a new unidentified ~130 kDa MIP (microtubule inner protein) at the inner junction, as shown in Figure 6. This new MIP is interesting because it connects between the inner junction and the further part of B-tubules.

In addition, the authors analyzed the structure of FAP115KO mutant DMT. The phenotype of this FAP115KO was recently analyzed and published by the authors' group. Therefore the structural analysis is a good complement for the analysis.

Overall, the paper is technically sound and a good advancement of the structural study in the cilia/flagella field. The reviewer recommends this paper for publication, with some minor revisions of texts and figures.

Major points:

(1) Please explain or discuss why the new ~130 kDa MIP at the inner junction was not found (or indicated as minor density) in the previous publications.

What percentage of the DMT were used to show the new MIP?

Response:

Although we are not certain the exact reason why the previous works were not able to identify the 130KDa MIP at the inner junction of DMT, we believe that extensive focused classification and refinement is the key to successful identifying new structure feature in our study.

We speculate that in previous work by using cryoEM and single particle analysis, it is challenging to separate signal from the microtubule and from the MIPs. Since the microtubule backbone composed of alpha/beta-tubulin was a dominant structure feature, it would interfere with the classification of 2D images focusing on MIPs, making identification of MIPs difficult. For this reason, excluding the signal contributed by the MT backbone is critical. Ma, *et al* 2019 did an excellent job in their study on the DMT structure from *Chlamydomonas*. By applying the multi-body refinement and density subtraction algorithm, focusing only on signals contributed by the MIPs, they were able to successfully classify MIPs, leading to a high resolution structure.

In our cryoET work, we have adapted a similar approach, by using focused 3D classification on MIPs in our subtomograms.

Below, we give an example of our 3D classification scheme. The 130KDa MIP near the inner junction was initially shown as a smear density in every 16nm. By using 3D classification on 5235 subtomograms, with a customized mask focusing on the inner junction, we were able to

identified six classes. Five classes, with 96% population, showed a MIP at three different locations that are 16 nm apart. By shifting these 5 classes, 5030 subtomograms, in-register followed by averaging, we identified a previous unknown ~130 KDa MIP at the inner junction.

We have emphasized this point in the “Material and Methods” section in page 58-59 of the manuscript.

[Figure removed by editorial staff per authors' request]

(2) Please show the difference between their T-DMT and the ones in the previous publication (probably in the supplementary figure) and discuss their technical advantages and/or disadvantages.

Response:

Due to the large size and multi-facet nature of the structure of the DMT, we think a detailed side-by-side comparisons of our structure to previous published results and a discussion on the technical advantages and disadvantages among methods used will be beyond the scope of this manuscript. Perhaps this merits another short article or assay in the future. Here, we provide a comparison of several structures, showing the seam region spanning protofilaments A06 to A10.

[Figure removed by editorial staff per authors' request]

As reviewer #2 has correctly pointed out, one of the main concerns for single particle cryoEM analysis (SPA) is the pre-treatment of sample, either by chemical or mechanical methods. This might disrupting the sample, resulting in partial or complete loss of the DMT components, such as MIPs. On the other hands, the SPA method benefits from sample embedded in relative thin ice. The data are in high signal-to-noise ratio (SNR), which can potentially lead to high resolution structure. In our cryoET study, the sample was kept in near-native state. The disruption to axonemal structure was kept at minimum. However, the sample was in relatively thick ice (250~200 nm), resulting in lower SNR and lower resolution than the SPA approach.

Minor points:

(1) Figure 4G does not clearly show the colors of dots, because they are too small. In addition, the near and far side dots of DMT are overlapping. Please consider a better presentation, such as an unfolded form of DMT.

Response:

We have now sent in a full-size figure in high-resolution in our revision.

(2) Table 1: "Number of subtomograms" should be "Number of subtomographic particles"

Response:

We'd prefer to use the term "subtomogram" as this is a commonly used term in the field.

November 26, 2021

RE: Life Science Alliance Manuscript #LSA-2021-01225-TR

Dr. Sam Li
University of California, San Francisco
Dept. of Biochemistry
Univ. California, San Francisco
600, 16th Street
San Francisco, California 94158

Dear Dr. Li,

Thank you for submitting your revised manuscript entitled "Electron Cryo-Tomography Structure of Axonemal Doublet Microtubule from *Tetrahymena thermophila*". We would be happy to publish your paper in Life Science Alliance pending final revisions necessary to meet our formatting guidelines.

- please add the twitter handle of your host institute/organization as well as your own or/and one of the authors in our system
- please consult our manuscript preparation guidelines <https://www.life-science-alliance.org/manuscript-prep> and make sure your manuscript sections are all there and in the correct order;
- please use the [10 author names, et al.] format in your references (i.e. limit the author names to the first 10)
- please upload one figure per file and be sure to label the panels in the figures
- please add your main, supplementary figure, and table legends to the main manuscript text after the references section
- all figure legends should only appear in the main manuscript file
- Tables can be included at the bottom of the main manuscript file or be sent as separate files.
- please provide a separate Data Availability section

A. FINAL FILES:

B. MANUSCRIPT ORGANIZATION AND FORMATTING:

Sincerely,

Reviewer #2 (Comments to the Authors (Required)):

The paper was properly revised and this work is appropriate for publication.

December 6, 2021

RE: Life Science Alliance Manuscript #LSA-2021-01225-TRR

Dr. Sam Li
University of California, San Francisco
Dept. of Biochemistry
Univ. California, San Francisco
600, 16th Street
San Francisco, California 94158

Dear Dr. Li,

Thank you for submitting your Research Article entitled "Electron Cryo-Tomography Structure of Axonemal Doublet Microtubule from *Tetrahymena thermophila*". It is a pleasure to let you know that your manuscript is now accepted for publication in Life Science Alliance. Congratulations on this interesting work.

DISTRIBUTION OF MATERIALS:

Again, congratulations on a very nice paper. I hope you found the review process to be constructive and are pleased with how the manuscript was handled editorially. We look forward to future exciting submissions from your lab.

Sincerely,
